# Factors associated with successful completion of outpatient parenteral antibiotic therapy in an area with a high prevalence of multidrug-resistant bacteria: 30-day hospital admission and mortality rates

Thais Cristina Garbelini Salles[1,2][☯]*, Santiago Grau Cerrato[3][‡], Tatiana Fiscina Santana[4], Eduardo Alexandrino Medeiros[1][☯]

1 Division of Infectious Diseases, Department of Internal Medicine, Paulista School of Medicine, Federal University of São Paulo, São Paulo, São Paulo, Brazil, 2 Hospital Santa Helena, Santo André, São Paulo, Brazil, 3 Autonomous University of Barcelona, Campus de la UAB, Bellaterra, Barcelona, Spain, 4 UnitedHealth Group, São Paulo, São Paulo, Brazil

☯ These authors contributed equally to this work.
‡ This author also contributed equally to this work.
* thais.salles@yahoo.com.br

## Abstract

### Objectives

To identify factors associated with hospital admission and mortality within the first 30 days after enrolment in an outpatient parenteral antimicrobial therapy (OPAT) program, also analysing adequacy of the treatment regimen and clinical outcomes.

### Patients and methods

This was a retrospective cohort study conducted between October 2016 and June 2017 in the state of São Paulo, Brazil. Variables related to hospital admission and mortality were subjected to bivariate analysis, and those with a $P<0.05$ were subjected to multivariate analysis as risk factors.

### Results

We evaluated 276 patients, of whom 80.5% were ≥60 years of age and 69.9% had more than one comorbidity. Of the patients evaluated, 41.3% had pneumonia and 35.1% had a urinary tract infection. The most common etiological agent, isolated in 18 (31.6%) cases, was *Klebsiella pneumoniae*, and 13 (72,2%) strains were carbapenem resistant. The OPAT was in accordance with the culture results in 76.6% of the cases and with the institutional protocols in 76.4%. The majority (64.5%) of the patients were not admitted, and a cure or clinical improvement was achieved in 78.6%. Multivariate analysis showed that, within the first 30 days after enrolment, the absence of a physician office visit was a predictor of hospital admission ($P<0.001$) and mortality ($P = 0.006$).

**Data Availability Statement:** All relevant data are within the manuscript and its Supporting Information files.

**Funding:** This work was supported by the Hospital Epidemiology Research Group of the Federal University of São Paulo, registered with the Brazilian Conselho Nacional de Desenvolvimento Científico e Tecnológico (CNPq, National Council for Scientific and Technological Development). The company United Health Group provided support in the form of salaries for authors T.C.G.S. and E.A.M, but did not have any additional role in the study design, data collection and analysis, decision to publish, or preparation of the manuscript. The specific roles of these authors are articulated in the 'author contributions' section.

**Competing interests:** The company United Health Group provided support in the form of salaries for authors T.C.G.S. and E.A.M. This does not alter our adherence to PLOS ONE policies on sharing data and materials.

## Conclusions

This study demonstrated the viability of OPAT in elderly patients with pulmonary or urinary tract infections in an area with a high prevalence of multidrug-resistant bacteria and that a post-discharge physician office visit is protective against hospital admission and mortality.

## Introduction

Outpatient parenteral antimicrobial therapy (OPAT) programs were initially described more than 40 years ago for the treatment of patients with cystic fibrosis [1]. Since then, their use has been widespread and OPAT guidelines have been published in various countries [2–4]. Despite the many advantages, the administration of antimicrobial agents outside the hospital can also lead to complications, such as worsening of the infectious process and hospitalisation of the patient. Although there have been some studies on the topic, the risk factors associated with mortality and admission among patients enrolled in OPAT programs have yet to be well established. The impact that having a physician office visit soon after discharge has on the rates of hospital admission and mortality has not been widely discussed.

The objective of this study was to analyse the characteristics and clinical outcomes of patients enrolled in an OPAT program, to identify factors associated with hospital admission and death within the first 30 days after enrolment. In addition, adequacy to the antimicrobial treatment regimens prescribed were evaluated in the context of an area with a high prevalence of multidrug-resistant bacteria.

## Patients and methods

This was a retrospective cohort study conducted between October 2016 and June 2017. The patients evaluated were among those treated by OPAT via the Santa Helena/Amil United-Health Group network, which comprises the 137-bed Santa Helena Hospital, a tertiary care hospital located in the city of Santo André, in the Brazilian state of São Paulo, as well as 12 out-patient clinics in adjacent municipalities.

The requests for inclusion in the OPAT program were made by the physicians, each of whom filled out a form with the data regarding the treatment, as well as the clinical and epidemiological characteristics of the patient in question. To be included in the program, patients were required to meet the following criteria: being clinically stable; having experienced no adverse events related to the first antimicrobial administration; having no history of psychiatric disorders or current drug addiction; having a cooperative family member or caregiver; having a contact telephone number; and having a means of transport. Patients who were candidates for oral antimicrobial treatment were excluded, as were those who were under 18 years of age, those who has previously undergone OPAT and those for whom no post-discharge physician office visit was scheduled. Also, if information regarding the outcome of the treatment was not reported, the patient was excluded from the study. To estimate 10-year mortality, the Charlson comorbidity score was calculated for each patient [5]. All comorbidities were registered according to each patient´s medical history and classified in conformity with the International Classification of Diseases (ICD) [6]. The antimicrobial agents were administered at one of the Santa Helena/Amil UnitedHealth Group network health care facilities or in the home of the patient. Patients who were bedridden and whose family did not have a means of transport to one of the facilities were assigned

antimicrobial treatment at home. The antibiotic at home was administered by the nursing staff assigned to visit the patient according to the antimicrobial scheme prescribed. All antibiotics were infused by gravity or *in bolus*.

The use of ceftriaxone, piperacilin-tazobactam, cefepime, amikacin, meropenem, teicoplanin and vancomycin was evaluated for the influence on hospital admission or mortality. Culture-guided adequacy of the OPAT regimen was defined as when the treatment was prescribed based on the results of the cultures. We determined whether each patient had visited a physician within the first 30 days after inclusion in the OPAT program, and we identified the reason for hospital admission (non-infectious disease, treatment failure or a new infection), as well as whether a cure or clinical improvement was achieved. Treatment failure was defined as the maintenance or worsening of an initial infectious condition, as evidenced by the appearance of new signs and symptoms or the exacerbation of existing symptoms.

Outcomes were classified as cure or clinical improvement; treatment failure; or death. The cure or clinical improvement outcome was defined as the resolution or improvement of an initial infectious condition, evidenced by the reduction or disappearance of signs and symptoms presented during the initial infectious process.

A database was constructed from October 2016 to June 2017 and accessed for the purpose of this study from October 2016 to July 2018. Patient data was obtained from the OPAT inclusion request forms and electronic medical records. The requirement for informed consent was waived by the research ethics committees. The variables were evaluated prospectively to determine factors related to hospital admission and death within the first 30 days after inclusion in the OPAT program.

In the statistical analysis, continuous variables were summarized as mean and standard deviation, whereas categorical variables were summarized as absolute and relative frequencies. To identify the factors that influenced hospital admission and mortality, a bivariate analysis of each factor was performed using the Student's *t*-test for independent samples (for associations of continuous variables) and the chi-square test (for associations of categorical variables). We performed an additional bivariate analysis excluding palliative care patients. In a subsequent multivariate analysis using multiple logistic regression models, the factors with a $P<0.05$ in the bivariate analysis were included as independent variables. We calculated odds ratios and the respective confidence intervals. Data processing and analysis were performed with the IBM SPSS Statistics software package, version 22.0 (IBM Corp., Armonk, NY, USA). Values of $P<0.05$ were considered statistically significant.

The project was approved by the Research Ethics Committees of the Federal University of São Paulo and of the Amil UnitedHealth Group. The requirement for informed consent was waived by both committees.

The patients and methods is available at doi.org/10.17504/protocols.io.bmuyk6xw.

## Results

There were 476 physician requests for inclusion of a patient in the OPAT program. Of those 476 records, 171 were excluded for not meeting the study criteria (Fig 1). Therefore, the final sample comprised 276 patients, corresponding to 1748 patient-days. Females accounted for 57.6% of the sample, and 80.5% of the patients in the sample were $\geq$ 60 years of age.

Of the 276 patients evaluated, 188 (69.9%) had two or more comorbidities and 37 (13.4%) were under palliative care. The most common types of infections were community-acquired infections, in 190 patients (69.9%), of which 114 (41.3%) were pneumonia and 97 (35.1%) were urinary tract infections. There was no association found between the type of infectious diagnosis and admission or death.

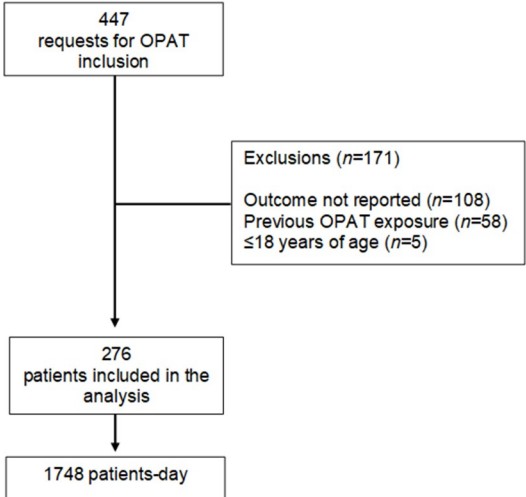

**Fig 1. Flow chart of the patient selection process for inclusion in an Outpatient Parenteral Antimicrobial Therapy (OPAT) program at facilities within the Santa Helena\Amil UnitedHealth Group network, October 2016 through June 2017.**

Most antibiotics (79,9%) were prescribed during hospitalization and the treatment was concluded as OPAT. The treatment was in accordance with the results of cultures in 36 cases (76.6%) and with the institutional protocols in 211 (76.4%). Of the 276 patients, 239 (86.6%) received monotherapy, 114 (36.4%) being treated with ceftriaxone. Peripheral venous catheters were used in 195 cases (79.6%).

Cultures were collected from 162 (58.7%) of the 276 patients, and bacteria were isolated from 57 (35.2%) of the cultures. The bacteria most commonly isolated from the positive cultures were *Klebsiella pneumoniae*, in 18 cases (31.6%), and *Escherichia coli*, in 14 (24.6%). Antimicrobial susceptibility test results were available for 47 (88.7%) of the positive cultures. Multidrug-resistant bacteria were isolated from 14 cultures (29.8%), and eight (57.1%) of the affected patients were classified as having a healthcare-associated infection. Of the multidrug-resistant bacteria isolated, one (7.1%) was vancomycin-resistant *Enterococcus faecalis* isolated from a culture of a skin secretion sample and 13 (92.9%) were carbapenem-resistant and aminoglycoside-susceptible *K. pneumoniae* isolated from cultures of urine samples. Most of these latter patients (69,2%) received microbiologically appropriate treatment with either amikacin or gentamicin and 77,8% had an outcome of cure or clinical improvement.

Of the 14 patients affected by multidrug-resistant bacteria, 11 (78.6%) were admitted to hospital, six (42.9%) because of treatment failure, and one (7.1%) died while under palliative care. Among the six (42.9%) patients who were admitted to hospital because of treatment failure, the OPAT was in line with the institutional protocols in three (50.0%) and was considered microbiologically appropriate in (50.0%).

Of the 276 patients evaluated, 98 (35.5%) were admitted to hospital within the first 30 days after inclusion in the OPAT program. Among those 98 patients, the reason for hospital admission was treatment failure in 35 (35.7%), a new infection in 22 (22.5%) and a non-infectious disease in 41 (41.8%).

A post-discharge physician office visit was documented in 74 (75.5%) of the 98 admitted patients. In the sample, a cure or clinical improvement was achieved in 217 patients (78.6%), treatment failure occurred in 33 (12.0%), and 26 (9.4%) died. In an additional analysis that

excludes patients on palliative care, cure or clinical improvement was achieved in 205 patients (85.8%), treatment failure occurred in 26 (10.9%), and 8 (3.3%) died.

## Factors associated with hospital admission

In the bivariate analysis, the only factors significantly associated with hospital admission were palliative care ($P<0.001$), culture-guided adequacy of the OPAT ($P = 0.049$) and post-discharge physician office visit ($P<0.001$) (Table 1). Cross-frequency analysis indicated that the proportion of admitted patients was higher among those who were receiving palliative care than among those who were not (67.6% vs. 31.0%). In addition, the proportion of patients admitted to hospital was lower among the patients in whom there was culture-guided adequacy of the OPAT than among those in whom there was not (38.9% vs. 72.7%), whereas it was higher among those who had a post-discharge physician office visit than among those who did not (60.9% vs. 27.1%).

The proportion of patients admitted to hospital was 50.0% among the patients with neoplasia, compared with 32.0% among those without, the difference being statistically significant ($P = 0.028$). None of the other pathologies or diagnoses evaluated were significantly associated with hospital admission.

The results of the multiple logistic regression model, in which the variables showing $P<0.05$ in the bivariate analysis were included as independent variables are presented below (Table 2). The factors found to be independent predictors of hospital admission in the multiple logistic regression were palliative care ($P = 0.003$), a post-discharge physician office visit ($P<0.001$) and culture-guided adequacy of the OPAT ($P = 0.055$). Being under palliative care was found to increase the risk of hospital admission by 96.4% (OR = 3.64; 95% CI: 1.56–8.49). In contrast, a post-discharge physician office visit was found to reduce the risk of hospital admission by 75% (OR = 0.25; 95% CI: 0.14–0.45), and culture-guided adequacy of the OPAT was found to reduce that risk by 80% (OR = 0.20; 95% CI: 0.04–1.04). Neoplasia was found to have no significant effect on the risk of hospital admission ($P = 0.447$).

## Factors associated with mortality

The factors significantly associated with mortality in the bivariate analysis were palliative care ($P<0.001$), a post-discharge physician office visit ($P = 0.002$) and neoplasia ($P = 0.05$) (Table 3). Cross-frequency analysis showed that the mortality rate was higher among the patients who received palliative care than among those who did not (48.6% vs. 3.3%), as well as being higher among the patients who had a post-discharge physician office visit than among those who did not (18.8% vs. 6.3%) and among those patients with neoplasia in comparison with those without (17.5% vs. 7.8%). None of the infectious diagnoses were significantly associated with mortality.

The results of the multiple logistic regression model, in which the variables showing $P<0.05$ in the bivariate analysis were included as independent variables are listed below (Table 4). The factors found to be independent predictors of mortality in the multiple logistic regression were palliative care ($P<0.001$) and a post-discharge physician office visit ($P = 0.051$). Being under palliative care was found to increase the risk of death by 70% (OR = 30.00; 95% CI: 9.68–92.93). In contrast, a post-discharge physician office visit was found to reduce the risk of death by 63% (OR = 0.37; 95% CI: 0.13–1.00). Neoplasia was found to have no significant effect on the risk of death ($P = 0.526$).

## Factors associated with hospital admission and mortality in the absence of palliative care

Of the 276 patients evaluated, 37 (13.4%) were under palliative care. Among the 239 (86.6%) patients not receiving palliative care, only a post-discharge physician office visit showed a

**Table 1. Bivariate analysis of factors associated with hospital admission among patients enrolled in an outpatient parenteral antimicrobial therapy program (*n* = 276) at facilities within the Santa Helena/Amil UnitedHealth Group network, October 2016 through June 2017.**

| Characteristic | Admission | | P |
|---|---|---|---|
| | **Yes** | **No** | |
| | **(*n* = 98)** | **(*n* = 178)** | |
| Gender | | | |
| female (*n* = 159) | 58 (36.5%) | 101 (63.5%) | 0.694[a] |
| male (*n* = 117) | 40 (34.2%) | 77 (65.8%) | |
| Age group | | | |
| 19–59 years (*n* = 54) | 37 (68.5%) | 17 (31.5%) | 0.560[a] |
| 60–79 years (*n* = 104) | 69 (66.3%) | 35 (33.7%) | |
| ≥80 years (*n* = 118) | 72 (61.0%) | 46 (39.0%) | |
| Age at enrolment (years), mean ± SD | 73.7 ± 17.6 | 71.3 ± 17.0 | 0.272[b] |
| Type of infection | | | |
| healthcare-associated infections (*n* = 82) | 29 (35.4%) | 53 (64.6%) | 0.881[a] |
| community-acquired (*n* = 190) | 69 (36.3%) | 121 (63.7%) | |
| Setting of antimicrobial prescription | | | |
| in-hospital (*n* = 220) | 79 (35.9%) | 141 (64.1%) | 0.442[a] |
| emergency room (*n* = 33) | 9 (27.3%) | 24 (72.7%) | |
| outpatient clinic/home (*n* = 23) | 10 (43.5%) | 13 (56.5%) | |
| Access | | | |
| peripheral venous catheter (*n* = 195) | 75 (38.5%) | 120 (61.5%) | 0.060[a] |
| short-term central venous catheter (*n* = 48) | 13 (27.1%) | 35 (72.9%) | |
| permanent or semi-permanent central venous catheter (*n* = 2) | 2 (100.0%) | 0 (0.0%) | |
| Days of treatment, mean ± SD | 6.1 ± 2.7 | 6.5 ± 7.1 | 0.645[b] |
| Adequacy to the protocol | | | |
| yes (*n* = 211) | 74 (35.1%) | 137 (64.9%) | 0.785[a] |
| no (*n* = 65) | 24 (36.9%) | 41 (63.1%) | |
| Culture-guided adequacy | | | |
| yes (*n* = 36) | 14 (38.9%) | 22 (61.1%) | 0.049[a] |
| no (*n* = 11) | 8 (72.7%) | 3 (27.3%) | |
| Palliative care | | | |
| yes (*n* = 37) | 25 (67.6%) | 12 (32.4%) | < 0.001[a] |
| no (*n* = 239) | 73 (30.5%) | 166 (69.5%) | |
| Sample collected for culture | | | |
| yes (*n* = 162) | 62 (38.3%) | 100 (61.7%) | 0.253[a] |
| no (*n* = 114) | 36 (31.6%) | 78 (68.4%) | |
| Physician office visit after enrolment | | | |
| yes (*n* = 207) | 56 (27.1%) | 151 (72.9%) | < 0.001[a] |
| no (*n* = 69) | 42 (60.9%) | 27 (39.1%) | |

SD, standard deviation; OPAT, outpatient parenteral antimicrobial therapy.
[a]chi-square test.
[b]Student's t-test.

statistically significant association with hospital admission (*P*<0.001), the proportion of admitted patients being lower among those who had a post-discharge office visit than among those who did not (22.7% vs. 57.4%). Similarly, a post-discharge physician office visit was the only factor showing a statistically significant association with mortality (*P* = 0.006), the mortality rate being lower among those who had a post-discharge office visit than among those who

**Table 2. Multivariate analysis of factors associated with hospital admission (independent variable) among patients enrolled in an outpatient parenteral antimicrobial therapy program at facilities within the Santa Helena/ Amil UnitedHealth Group network, October 2016 through June 2017.**

| Factor (independent variable) | OR | 95% CI | P |
|---|---|---|---|
| Palliative care | 3.64 | 1.56–8.49 | 0.003 |
| Physician office visit after enrolment | 0.25 | 0.14–0.45 | < 0.001 |
| Neoplasia | 0.45 | 0.63–3.00 | 0.447 |
| Culture-guided adequacy | 0.20 | 0.04–1.04 | 0.055 |

OPAT, outpatient parenteral antimicrobial therapy.

did not (1.6% vs. 9.3%). Multivariate analysis excluding palliative care was not performed, given that only one variable showed a statistically significant association with hospital admission and mortality.

## Discussion

The OPAT programs are internationally recognised as an option for the treatment of a variety of infectious diseases and their use is expected to grow exponentially in the coming years. Those characteristics led to the development of the present study. In Brazil, "hospital at home" programs started in the 1960s and from the 1990s onwards, the concept of home care as a modality evolved, with the inclusion of multi-professional teams. In 2011, the "Best at Home" program was launched by the Public Health System, to allow home care for those with temporary or permanent mobility difficulties. The OPAT programs emerged in the following years and in 2017, the Brazilian Society of Infectious Diseases published national guidelines to promote the expansion of this treatment modality in the country [3].

Most of the studies analysing the efficacy and safety of OPAT regimens have been retrospective cohort studies [7–9]. There have been few studies of the use of OPAT in Brazil. One study assessed the use of OPAT, within the context of the Brazilian Unified Health Care System (universal health care), at a university hospital specialising in orthopaedics, reporting the training of 450 professionals and low admission rates for the patients enrolled in the OPAT program [10]. However, the authors of that study did not evaluate the outcomes of the patients evaluated. In 2018, Psaltikidis et al performed an analysis that demonstrated the cost-effectiveness of OPAT in the Brazilian Unified Health Care System [11].

The experience with OPAT is also limited in other Latin American countries. In a study conducted in Argentina, Lopardo et al. evaluated 48 cases of endocarditis treated with OPAT in different contexts: at infusion centres, via home-based nursing and by self-administration and concluded that OPAT can safely and effectively treat selected patients with endocarditis when adequate selection criteria are used [12]. In Chile, clinical outcomes and OPAT-related costs were analysed in a prospective cohort of 111 children with urinary tract infections and compared with those observed for a group of 81 hospitalised children. OPAT was equally effective, safer and significantly less expensive than inpatient care [13]. In Mexico, outpatient ertapenem therapy was studied in an ESBL-high-prevalence area and the treatment course was found to be effective, safe and cost-saving when compared to inpatient care [14].

Of the 276 patients in our cohort, 118 (42.8%) were over 80 years of age. A comparative study of young and elderly patients undergoing OPAT has shown that it is safe and efficacious in the elderly group, with rates of clinical improvement above 92% [15].

The main infections treated with OPAT in the present study were pneumonia and urinary tract infections, with a predominance of community-acquired infections. Our data differ from

**Table 3. Bivariate analysis of factors associated with mortality among patients enrolled in an outpatient parenteral antimicrobial therapy program (*n* = 276) at facilities within the Santa Helena/Amil UnitedHealth Group network, October 2016 through June 2017.**

| Characteristic | Death | | P |
|---|---|---|---|
| | Yes | No | |
| | (*n* = 26) | (*n* = 250) | |
| Gender | | | |
| female (*n* = 159) | 18 (11.3%) | 141 (88.7%) | 0.208[a] |
| male (*n* = 117) | 8 (6.8%) | 109 (93.2%) | |
| Age group | | | |
| 19–59 years (*n* = 54) | 51 (94.4%) | 3 (5.6%) | 0.553[a] |
| 60–79 years (*n* = 104) | 93 (89.4%) | 11 (10.6%) | |
| ≥80 years (*n* = 118) | 106 (89.8%) | 12 (10.2%) | |
| Age at enrolment (years), mean ± SD | 75.3 (15.2) | 71.9 (17.4) | 0.337[b] |
| Type of infection | | | |
| healthcare-associated infections (*n* = 82) | 8 (9.8%) | 74 (90.2%) | 0.942[a] |
| community-acquired (*n* = 190) | 18 (9.5%) | 172 (90.5%) | |
| Setting of antimicrobial prescription | | | |
| in-hospital (*n* = 220) | 19 (8.6%) | 201 (91.4%) | 0.486[a] |
| emergency room (*n* = 33) | 5 (15.2%) | 28 (84.8%) | |
| outpatient clinic/home (*n* = 23) | 2 (8.7%) | 21 (91.3%) | |
| Access | | | |
| peripheral venous catheter (*n* = 195) | 17 (8.7%) | 178 (91.3%) | 0.652[a] |
| short-term central venous catheter (*n* = 48) | 6 (12.5%) | 42 (87.5%) | |
| permanent or semi-permanent central venous catheter (*n* = 2) | 0 (0.0%) | 2 (100.0%) | |
| Days of treatment, mean ± SD | 5.3 (2.4) | 6.4 (6.1) | 0.353[b] |
| Adequacy to the protocol | | | |
| yes (*n* = 211) | 19 (9.0%) | 192 (91.0%) | 0.670[a] |
| no (*n* = 65) | 7 (10.8%) | 58 (89.2%) | |
| Culture-guided adequacy | | | |
| yes (*n* = 36) | 2 (5.6%) | 34 (94.4%) | 0.675[a] |
| no (*n* = 11) | 1 (9.1%) | 10 (90.9%) | |
| Palliative care | | | |
| yes (*n* = 37) | 18 (48.6%) | 19 (51.4%) | < 0.001[a] |
| no (*n* = 239) | 8 (3.3%) | 231 (96.7%) | |
| Sample collected for culture | | | |
| yes (*n* = 162) | 15 (9.3%) | 147 (90.7%) | 0.913[a] |
| no (*n* = 114) | 11 (9.6%) | 103 (90.4%) | |
| Physician office visit after enrolment | | | |
| yes (*n* = 207) | 13 (6.3%) | 194 (93.7%) | 0.002[a] |
| no (*n* = 69) | 13 (18.8%) | 56 (81.2%) | |

SD, standard deviation; OPAT, outpatient parenteral antimicrobial therapy.

[a]chi-square test.

[b]Student's t-test.

those reported in other studies, in which skin, soft tissue, and osteoarticular infections predominated [16,17].

Multidrug-resistant bacteria were isolated in 29.8% of the positive cultures. There has been an increase in the emergence of carbapenem-resistant strains of Enterobacteriaceae, in Brazil

**Table 4. Multivariate analysis of factors associated with mortality (independent variable) among patients enrolled in an outpatient parenteral antimicrobial therapy program at facilities within the Santa Helena/Amil United-Health Group network, October 2016 through June 2017.**

| Factor (independent variable) | OR | 95% CI | *P* |
|---|---|---|---|
| Palliative care | 3.64 | 1.56–8.49 | 0.003 |
| Physician office visit after enrolment | 0.25 | 0.14–0.45 | < 0.001 |
| Neoplasia | 0.45 | 0.63–3.00 | 0.447 |

and worldwide [18]. Mazzariol et al. demonstrated a global trend of emergence of multidrug-resistant extended-spectrum beta-lactamase–producing strains of Enterobacteriaceae in urinary tract infections [19]. The authors found that infection with such strains did not represent a risk factor for mortality or hospital admission, differing from what has been reported in other studies that evaluated those outcomes in patients with multidrug-resistant bacterial infection [20,21]. This difference might be explained by the fact that in this study, most patients with multidrug-resistant isolates had urinary tract infections caused by K. pneumoniae strains that were susceptible to aminoglycosides, which were available for targeted short-term treatment in OPAT.

A cure or clinical improvement was achieved in the great majority of the patients evaluated in the present study. Barr et al. conducted a 10-year retrospective cohort study of an OPAT program at a teaching hospital in the United Kingdom [7]. The authors observed a predominance of skin and soft tissue infections and reported that a cure or clinical improvement was achieved in 92.4% of the 2233 patients evaluated. Data from our study might indicate that there was inappropriate selection and follow-up of the patients included in the OPAT program, demonstrating the importance of the involvement of a multidisciplinary team comprising an infectious diseases specialist who can identify eligible patients and provide the necessary support.

Multivariate analysis showed only one significant risk factor for hospital admission and mortality: not having had a post-discharge physician office visit within the first 30 days after inclusion in the OPAT program. A consultation after the initiation of OPAT allows test results to be assessed, antimicrobial- and infusion-related adverse events to be identified, and treatment to be tailored according to the culture results, if they are still pending. Jencks et al. analysed data from 11.855.702 patients hospitalised in the United States health care system and showed that 50.2% of the patients admitted within the first 30 days after discharge had not had a physician office visit between discharge and hospital admission [22]. The main OPAT guidelines recommend clinical evaluation of the patient after inclusion in the program [2,4]. However, there is no consensus on how often those evaluations should occur. Palms et al. developed a 30-day admission predictive model for patients discharged to OPAT and found that follow-up at a clinic specialising in infectious diseases was a preventive factor for hospital admission [23]. Saini et al. demonstrated that evaluation by an infectious diseases specialist within the first 14 days after enrolment in an OPAT program is associated with a lower risk of hospital admission within the first 30 days after discharge [24]. To our knowledge, there have been no studies establishing a direct relationship between post-enrolment evaluation and mortality.

Our study has some limitations. First, we did not perform a patient satisfaction analysis, which could provide important information about how well the OPAT program works. In addition, we did not perform a cost analysis of the program. However, the 1748 patient-days on OPAT can translate to an equal number of inpatient bed-days saved. Despite these limitations, we have shown that OPAT treatments can be a substitute model for elderly patients who are clinically stable and appropriately selected for inclusion in the program.

Future studies should focus on the use of OPAT programs where there is limited access to health care services, such as in rural and riparian communities. In addition, the potential for expanding such programs using telemedicine should be evaluated.

## Supporting information

**S1 File.**
(XLSX)

## Acknowledgments

We are grateful to the administration of the Santa Helena Hospital, as well as to the staff of the Amil UnitedHealth Group, for their support.

## Author Contributions

**Conceptualization:** Thais Cristina Garbelini Salles.

**Project administration:** Eduardo Alexandrino Medeiros.

**Supervision:** Santiago Grau Cerrato, Eduardo Alexandrino Medeiros.

**Validation:** Santiago Grau Cerrato, Tatiana Fiscina Santana.

**Writing – original draft:** Thais Cristina Garbelini Salles.

**Writing – review & editing:** Thais Cristina Garbelini Salles.

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
