## [Decision Letter · Decision Letter 0]

21 Sep 2020

PONE-D-20-18208

Factors associated with successful completion of outpatient parenteral antibiotic therapy in an area with a high prevalence of multidrug-resistant bacteria: 30-day hospital admission and mortality rates

PLOS ONE

Dear Dr.ssa Salles,

Thank you for submitting your manuscript to PLOS ONE. After careful consideration, we feel that it has merit but does not fully meet PLOS ONE’s publication criteria as it currently stands. Therefore, we invite you to submit a revised version of the manuscript that addresses the points raised during the review process.

We look forward to receiving your revised manuscript.

Kind regards,

Francesco Di Gennaro

Academic Editor

PLOS ONE

Journal Requirements:

2. In your ethics statement in the Methods section and in the online submission form, please provide additional information about the data used in your retrospective study. Specifically, please ensure that you have discussed whether all data were fully anonymized before you accessed them and/or whether the IRB or ethics committee waived the requirement for informed consent. If patients or next of kin provided informed written consent to have data from their medical records used in research, please include this information.

3. Please clarify what date range the patient medical records were originally recorded and what date(s) you accessed the medical records.

4.Thank you for stating the following in the Financial Disclosure section:

[This work was granted to the corresponding author, T.C.G.S. and was supported by the Hospital Epidemiology Research Group of the Federal University of São Paulo, registered with the Brazilian Conselho Nacional de Desenvolvimento Científico e Tecnológico (CNPq, National Council for Scientific and Technological Development), URL http://www.cnpq.br/. No role was played in the study design, data collection, analysis, decision to publish or preparation of the manuscript.].   

We note that one or more of the authors are employed by a commercial company: UnitedHealth Group

Additional Editor Comments (if provided):

Dear Authors,

follow reviewer suggestion to improve your article

Reviewers' comments:

Reviewer's Responses to Questions

**Comments to the Author**

1. Is the manuscript technically sound, and do the data support the conclusions?

Reviewer #1: Partly

Reviewer #2: Yes

2. Has the statistical analysis been performed appropriately and rigorously? 

Reviewer #1: Yes

Reviewer #2: Yes

3. Have the authors made all data underlying the findings in their manuscript fully available?

Reviewer #1: Yes

Reviewer #2: Yes

4. Is the manuscript presented in an intelligible fashion and written in standard English?

Reviewer #1: Yes

Reviewer #2: Yes

5. Review Comments to the Author

Reviewer #1: PLOS ONE

Full Tittle: Factors associated with successful completion of outpatient parenteral antibiotic therapy in an area with a high prevalence of multidrug-resistant bacteria: 30-day hospital admission and mortality rates

Manuscript Number: PONE-D-20-18208

Article Type: Research Article

Corresponding Author: Thais Cristina Garbelini Salles, M.D. Universidade Federal de São Paulo – UNIFESP. Sao Paulo, SP BRAZIL

QUESTIONS/SUGGESTIONS

● PATIENTS AND METHODS:

❖ Does the fact that a patient is assigned the antibiotic in the Santa Helena/Amil United Health Group network or at the patient's home, respond to any special criteria?

❖ It is important to know who administered the antibiotic at home: the nursing staff, the patients or their caregiver. How was the visiting schedule of nursing staff at the patient's home? What was the method of administration of the antibiotic? Was it by gravity or by the use of an elastomeric or electronic pump? Could be the type of administration of the antibiotic related to a higher mortality rate or a higher hospital admission?

❖ LINE 102, “The variables were evaluated prospectively to determine factors related to hospital admission”. Reading the article, the conclusion reached is that the study is retrospective, based on a later analysis of a group of data extracted from daily clinical practice. Could the authors clarify this point?

● RESULTS:

❖ Do the authors dispose of Charlson’s Index of the patients?, Could the Charlson index be related to a higher mortality rate or to a higher hospital admission?

❖ If data were available, it would be important to know what the referring department of the patients (medical specialties, surgical specialties, emergency department) was. Was the referring department of patients associated with greater therapeutic failure, mortality or hospital admission? In case that patients had been previously admitted to the hospital, ¿what was the average stay in conventional hospitalization? What was the duration of the antibiotic treatment during the conventional hospitalization? If the requested data were included, the methods section should also be modified.

❖ Another piece of information to provide, if available, would be to classify pneumonia according to FINE or CURB-65 severity index. Have FINE or CURB-65 any relationship with mortality or hospital admission?

❖ What other antibiotics were used besides ceftriaxone? Is any antibiotic associated with a higher percentage of hospital admissions or mortality?

❖ Line 142, “of the MDR bacteria, 13 (92,9%) were carbapenem-resistant K.pneumoniae…” What type of carbapenamase did the authors find? What antibiotics did they use in those cases?

● DISCUSSION:

❖ In recent years, Hospital at home (HaH) programs have spread throughout all the world. In these programs, medical and nursing staff administer hospital-level care at patient’s homes avoiding admission to a hospital ward. OPAT is one of the most frequent modalities in HaH programs with excellent results. I suggest that authors consider and comment about these HaH programs in the discussion section because this would reinforce the importance of medical visits in the OPAT programs.

● COMMENTS TO THE AUTHOR:

❖ The article draws attention to the high mortality of the series (9.4%) and the high percentage of hospital admissions (35.5%) within 30 days after inclusion in the OPAT program in comparison with other studies on OPAT in which clinical control by nurses and medical staff is carried out at the patient's home several times during the admission and with much better results. On the other hand, we must consider that part of the results are explained because study population is elderly and most of the infections are respiratory, so medical visits are essential.

❖ Personally, I think some data is lacking to give it consistency. All new requested data, if available, would greatly complement the information of the population under study and should be emphasized in the discussion of results.

Reviewer #2: In this interesting and innovative study, Authors investigated outcome and predictors of failure of outpatient parenteral antibiotic therapy. Interestingly, in this study the infectious diseases specialist consultation within the first 14 days after enrolment in an OPAT program is associated with a lower risk of hospital admission and mortality. Similarly, other important finding is the high efficacy of OPAT in treating community mild/moderate infections.

Overall, this work is worth for publication, however some revisions are needed:

1) A more detailed description of comorbidities should be given. For instance, the variable “neoplasia” presented in Table 2 was chosen arbitrarily? What about chronic kidney diseases, diabetes, immunesuppressive therapy etc? All of these factors could be associated with hospital admission

2) A more detailed description of treatment prescribed should be given, especially for multidrug resistant pathogens. Indeed, one of the main risk factors of failure is inappropriate treatment.

3) Mortality was quite low. Does it was associated with the infectious disease? The same description should be given regarding cause of hospital admission, if possible.

4) “The authors found that infection with such strains did not represent a risk factor for mortality or hospital admission” This statement should be better discussed. I suggest to describe more deeply the type of infection caused by MDR bacteria treated with outpatient treatment.

6. PLOS authors have the option to publish the peer review history of their article (what does this mean?). If published, this will include your full peer review and any attached files.

Reviewer #1: No

Reviewer #2: No

---

## [Author Response · Author response to Decision Letter 0]

4 Oct 2020

1. In your ethics statement in the Methods section and in the online submission form, please provide additional information about the data used in your retrospective study. Specifically, please ensure that you have discussed whether all data were fully anonymized before you accessed them and/or whether the IRB or ethics committee waived the requirement for informed consent. If patients or next of kin provided informed written consent to have data from their medical records used in research, please include this information.

The ethics committees waived the requirement for informed consent. This information was added in ethics statement and the patient methods section.

2. Please clarify what date range the patient medical records were originally recorded and what date(s) you accessed the medical records.

A database was constructed from October 2016 to June 2017 and accessed for the purpose of this study from October 2017 to July 2018. This information was included in the patients and methods section. 

3.Thank you for stating the following in the Financial Disclosure section:

[This work was granted to the corresponding author, T.C.G.S. and was supported by the Hospital Epidemiology Research Group of the Federal University of São Paulo, registered with the Brazilian Conselho Nacional de Desenvolvimento Científico e Tecnológico (CNPq, National Council for Scientific and Technological Development), URL http://www.cnpq.br/. No role was played in the study design, data collection, analysis, decision to publish or preparation of the manuscript.]. 

We note that one or more of the authors are employed by a commercial company: UnitedHealth Group

The updated Funding Statement was included in the cover letter with the information that the company United Health Group provided support in the form of salaries for the authors T.C.G.S and E.A.M. and did not play a role in the study design, data collection and analysis, decision to publish, or preparation of the manuscript. 

2. Please also provide an updated Competing Interests Statement declaring this commercial affiliation along with any other relevant declarations relating to employment, consultancy, patents, products in development, or marketed products,etc. 

The updated Funding Statement and Competing Interests Statement was included in the cover letter with the following informations:

• “The company United Health Group provided support in the form of salaries for the authors T.C.G.S and E.A.M. and did not play a role in the study design, data collection and analysis, decision to publish, or preparation of the manuscript. This does not alter our adherence to PLOS ONE policies on sharing data and materials.”

Review Comments

1. Is the manuscript technically sound, and do the data support the conclusions?

Reviewer #1: Partly

Reviewer #2: Yes

2. Has the statistical analysis been performed appropriately and rigorously?

Reviewer #1: Yes

Reviewer #2: Yes

3. Have the authors made all data underlying the findings in their manuscript fully available?

Reviewer #1: Yes

Reviewer #2: Yes

4. Is the manuscript presented in an intelligible fashion and written in standard English?

Reviewer #1: Yes

Reviewer #2: Yes

5. Review Comments to the Author

PATIENTS AND METHODS:

❖ Does the fact that a patient is assigned the antibiotic in the Santa Helena/Amil United Health Group network or at the patient's home, respond to any special criteria?

Patients who were bedridden and whose family did not have a means of transport to one of the facilities were assigned antimicrobial treatment at home. 

❖ It is important to know who administered the antibiotic at home: the nursing staff, the patients or their caregiver. How was the visiting schedule of nursing staff at the patient's home? What was the method of administration of the antibiotic? Was it by gravity or by the use of an elastomeric or electronic pump? Could be the type of administration of the antibiotic related to a higher mortality rate or a higher hospital admission?

The antibiotic at home was administered by the nursing staff assigned to visit the patient according to the antimicrobial scheme prescribed. All antibiotics were infused by gravity or in bolus. No analysis was made as to whether the type of infusion was related to higher mortality or hospital admission rates. However, no serious infusion-related adverse events were reported during this study.

❖ LINE 102, “The variables were evaluated prospectively to determine factors related to hospital admission”. Reading the article, the conclusion reached is that the study is retrospective, based on a later analysis of a group of data extracted from daily clinical practice. Could the authors clarify this point?

It is a retrospective study but the database was evaluated prospectively from the beginning of the OPAT program on October 2016. 

● RESULTS:

❖ Do the authors dispose of Charlson’s Index of the patients?, Could the Charlson index be related to a higher mortality rate or to a higher hospital admission?

Information regarding the patient´s Charlson Index was calculated, with a median score was 4 (four). These results were not analysed for the association with mortality or hospital admission rates.

❖ If data were available, it would be important to know what the referring department of the patients (medical specialties, surgical specialties, emergency department) was. Was the referring department of patients associated with greater therapeutic failure, mortality or hospital admission? In case that patients had been previously admitted to the hospital, ¿what was the average stay in conventional hospitalization? What was the duration of the antibiotic treatment during the conventional hospitalization? If the requested data were included, the methods section should also be modified.

Information on whether the antibiotic was prescribed in-hospital, at the emergency room, at an outpatient clinic or at a patinet´s home was submitted to a bivariate analysis and no statistically significant result was found concerning hospital admission and death (tables 1 and 3 of the manuscript). Most antibiotics (79,9%) were prescribed during hospitalization and the treatment was concluded as OPAT. In these cases, the average stay in conventional hospitalization or the duration of treatment was not calculated. 

❖ Another piece of information to provide, if available, would be to classify pneumonia according to FINE or CURB-65 severity index. Have FINE or CURB-65 any relationship with mortality or hospital admission?

Patients with pneumonia were not classified according to FINE or CURB-65 severity indexes. However, as an inclusion criteria, all patients were evaluated for clinical stability and only mild to moderate cases of pneumonia received OPAT. There was no association found between the type of infectious diagnosis and admission or death (tables 3 and 4).

❖ What other antibiotics were used besides ceftriaxone? Is any antibiotic associated with a higher percentage of hospital admissions or mortality?

A bivariate analysis of the Student’s t-test was applied to each antibiotic prescribed and the results did not show any influence on hospital admission or mortality, (Tables 1 and 2). The antibiotics evaluated were as follows: ceftriaxone, piperacilin-tazobactam, cefepime, amikacin, meropenem, teicoplanin and vancomycin. 

Table 1. Bivariate analysis of factors associated with hospital admission among patients enrolled in an outpatient parenteral antimicrobial therapy program (n=276) at facilities within the Santa Helena/Amil UnitedHealth Group network, October 2016 through June 2017

Antibiotic ADMISSION P

 Yes No 

Number of antibiotics prescribed 

Monotherapy (n = 239) 87 (36.4%) 152 (63.6%) 0.430

Combined therapy (n = 37) 11 (29.7%) 26 (70.3%) 

CEFTRIAXONE 

No (n = 162) 60 (37.0%) 102 (63.0%) 0.527

Yes (n = 114) 38 (33.3%) 76 (66.7%) 

PIPERACILIN-TAZOBACTAM 

No (n = 233) 83 (35.6%) 150 (64.4%) 0.926

Yes (n = 43) 15 (34.9%) 28 (65.1%) 

CEFEPIME 

No (n = 238) 81 (34.0%) 157 (66.0%) 0.200

Yes (n = 38) 17 (44.7%) 21 (55.3%) 

AMIKACIN 

No (n = 248) 85 (34.3%) 163 (65.7%) 0.203

Yes (n = 28) 13 (46.4%) 15 (53.6%) 

MEROPENEM 

No (n = 252) 91 (36.1%) 161 (63.9%) 0.497

Yes (n = 24) 7 (29.2%) 17 (70.8%) 

TEICOPLANIN 

No (n = 257) 95 (37.0%) 162 (63.0%) 0.063

Yes (n = 19) 3 (15.8%) 16 (84.2%) 

VANCOMYCIN 

No (n = 260) 93 (35.8%) 167 (64.2%) 0.714

Yes (n = 16) 5 (31.3%) 11 (68.8%) 

Table 2. Bivariate analysis of factors associated with mortality among patients enrolled in an outpatient parenteral antimicrobial therapy program (n=276) at facilities within the Santa Helena/Amil UnitedHealth Group network, October 2016 through June 2017

Antibiotic Death P

 Yes No 

Number of antibiotics prescribed 

Monotherapy (n = 239) 214 (89.5%) 25 (10.5%) 0.133

Combined therapy (n = 37) 36 (97.3%) 1 (2.7%) 

CEFTRIAXONE 

No (n = 162) 151 (93.2%) 11 (6.8%) 0.075

Yes (n = 114) 99 (86.8%) 15 (13.2%) 

PIPERACILIN-TAZOBACTAM 

No (n = 233) 211 (90.6%) 22 (9.4%) 0.977

Yes (n = 43) 39 (90.7%) 4 (9.3%) 

CEFEPIME 

No (n = 238) 215 (90.3%) 23 (9.7%) 0.729

Yes (n = 38) 35 (92.1%) 3 (7.9%) 

AMIKACIN 

 No (n = 248) 223 (89.9%) 25 (10.1%) 0.264

Yes (n = 28) 27 (96.4%) 1 (3.6%) 

MEROPENEM 

No (n = 252) 229 (90.9%) 23 (9.1%) 0.589

Yes (n = 24) 21 (87.5%) 3 (12.5%) 

TEICOPLANIN 

No (n = 257) 231 (89.9%) 26 (10.1%) 0.145

Yes (n = 19) 19 (100.0%) 0 (0.0%) 

VANCOMYCIN 

No (n = 260) 235 (90.4%) 25 (9.6%) 0.655

Yes (n = 16) 15 (93.8%) 1 (6.3%) 

❖ Line 142, “of the MDR bacteria, 13 (92,9%) were carbapenem-resistant K.pneumoniae…” What type of carbapenamase did the authors find? What antibiotics did they use in those cases?

The carbapenem-resistant K. pneumoniae samples were not analysed for the type of carbapenemase. These cultures were obtained from urine samples and 69,2% of the patients received microbiologically appropriate treatment with aminoglycosides.

● DISCUSSION:

❖ In recent years, Hospital at home (HaH) programs have spread throughout all the world. In these programs, medical and nursing staff administer hospital-level care at patient’s homes avoiding admission to a hospital ward. OPAT is one of the most frequent modalities in HaH programs with excellent results. I suggest that authors consider and comment about these HaH programs in the discussion section because this would reinforce the importance of medical visits in the OPAT programs.

The OPAT programs are internationally recognised as an option for the treatment of a variety of infectious diseases and their use is expected to grow exponentially in the coming years. In Brazil, “hospital at home” programs started in the 1960s and from the 1990s onwards, the concept of home care as a modality evolved, with the inclusion of multi-professional teams. In 2011, the "Best at Home" program was launched by the Public Health System, to allow home care for those with temporary or permanent mobility difficulties. The OPAT programs emerged in the following years and in 2017, the Brazilian Society of Infectious Diseases published national guidelines to promote the expansion of this treatment modality in the country.3 Those characteristics led to the development of the present study.

● COMMENTS TO THE AUTHOR:

❖ The article draws attention to the high mortality of the series (9.4%) and the high percentage of hospital admissions (35.5%) within 30 days after inclusion in the OPAT program in comparison with other studies on OPAT in which clinical control by nurses and medical staff is carried out at the patient's home several times during the admission and with much better results. On the other hand, we must consider that part of the results are explained because study population is elderly and most of the infections are respiratory, so medical visits are essential.

❖ Personally, I think some data is lacking to give it consistency. All new requested data, if available, would greatly complement the information of the population under study and should be emphasized in the discussion of results.

It is importante to point out that in an additional analysis that excludes patients on palliative care, the mortality rate is 3.3%; cure or clinical improvement was achieved in 205 patients (85.8%) and treatment failure occurred in 26 (10.9%). In both groups, the absence of a physician office visit was a predictor of hospital admission (P<0.001) and mortality (P=0.006).

Reviewer #2: In this interesting and innovative study, Authors investigated outcome and predictors of failure of outpatient parenteral antibiotic therapy. Interestingly, in this study the infectious diseases specialist consultation within the first 14 days after enrolment in an OPAT program is associated with a lower risk of hospital admission and mortality. Similarly, other important finding is the high efficacy of OPAT in treating community mild/moderate infections.

Overall, this work is worth for publication, however some revisions are needed:

1) A more detailed description of comorbidities should be given. For instance, the variable “neoplasia” presented in Table 2 was chosen arbitrarily? What about chronic kidney diseases, diabetes, immunesuppressive therapy etc? All of these factors could be associated with hospital admission.

The comorbidities studied were not chosen by the author arbitrarily as they were registered according to the medical history of the elligible OPAT patients. All commorbiditites were evaluated as independent variables and submitted to a bivariate analysis. Neoplasia appears in the multivariate analysis as it showed statistical significance (P<0.05) in the bivariate analysis for hospital admission and mortality (Tables 3 and 4). All commorbiditites were classified according to the International Classification of Diseases (ICD).

Table 3. Association of hospital admission with comorbidities and diagnosis – bivariate analysis, Santa Helena/Amil UnitedHealth Group network, October 2016 through June 2017

 ADMISSION P

 Yes No 

Comorbidities (1) 

Number of Comorbidities 

None or one (n = 81) 25 (30.9%) 56 (69.1%) 0.357

Two or more (n = 188) 69 (36.7%) 119 (63.3%) 

Circulatory system disease 

Yes (n = 202) 69 (34.2%) 133 (65.8%) 0.755

No (n = 69) 25 (36.2%) 44 (63.8%) 

Nutritional and metabolic endocrine disease 

Yes (n = 66) 24 (36.4%) 42 (63.6%) 0.742

No (n = 205) 70 (34.1%) 135 (65.9%) 

Neoplasia 

Yes (n = 40) 20 (50.0%) 20 (50.0%) 0.028

No (n = 231) 74 (32.0%) 157 (68.0%) 

Disease of the genitourinary system 

Yes (n = 31) 14 (45.2%) 17 (54.8%) 0.193

No (n = 240) 80 (33.3%) 160 (66.7%) 

Respiratory system disease 

Yes (n = 31) 13 (41.9%) 18 (58.1%) 0.368

No (n = 240) 81 (33.8%) 159 (66.3%) 

Nervous system disease 

Yes (n = 10) 4 (40.0%) 6 (60.0%) 0.719

No (n = 261) 90 (34.5%) 171 (65.5%) 

Infectious diagnosis (2) 

Pneumonia (n = 114) 36 (31.6%) 78 (68.4%) 

Urinary infection (n = 97) 39 (40.2%) 58 (59.8%) 0.411

Skin/soft tissue infection (n = 39) 13 (33.3%) 26 (66.7%) 

(1) The association with other comorbidities was not studied due to the reduced number of cases

(2) Other diagnoses were not included due to the reduced number of cases

Table 4. Association of death with comorbidities and diagnosis - bivariate analysis, Santa Helena/Amil UnitedHealth Group network, October 2016 through June 2017

 DEATH P

 Yes No 

Comorbidities (1) 

Number of Comorbidities 

None or one (n = 81) 8 (9.9%) 73 (90.1%) 0.829

Two or more (n = 188) 17 (9.0%) 171 (91.0%) 

Circulatory system disease 

Yes (n = 202) 19 (9.4%) 183 (90.6%) 0.860

Não (n = 69) 6 (8.7%) 63 (91.3%) 

Nutritional and metabolic endocrine disease 

Yes (n = 66) 6 (9.1%) 60 (90.9%) 0.965

Não (n = 205) 19 (9.3%) 186 (90.7%) 

Neoplasia 

Yes (n = 40) 7 (17.5%) 33 (82.5%) 0.050

Não (n = 231) 18 (7.8%) 213 (92.2%) 

Disease of the genitourinary system 

Yes (n = 31) 5 (16.1%) 26 (83.9%) 0.158

Não (n = 240) 20 (8.3%) 220 (91.7%) 

Respiratory system disease 

Yes (n = 31) 3 (9.7%) 28 (90.3%) 0.926

Não (n = 240) 22 (9.2%) 218 (90.8%) 

Nervous system disease 

Yes (n = 10) 1 (10.0%) 9 (90.0%) 0.931

Não (n = 261) 24 (9.2%) 237 (90.8%) 

Infectious diagnosis (2) 

Pneumonia (n = 114) 13 (11.4%) 101 (88.6%) 0.582

Urinary infection (n = 97) 7 (7.2%) 90 (92.8%) 

Skin/soft tissue infection (n =39) 4 (10.3%) 35 (89.7%) 

(1) The association with other comorbidities was not studied due to the reduced number of cases

(2) Other diagnoses were not included due to the reduced number of cases

2) A more detailed description of treatment prescribed should be given, especially for multidrug resistant pathogens. Indeed, one of the main risk factors of failure is inappropriate treatment.

A bivariate analysis of the Student’s t-test was applied to each antibiotic prescribed and the results did not show any influence on hospital admission or mortality, (Tables 1 and 2 of this review). The antibiotics evaluated were as follows: ceftriaxone, piperacilin-tazobactam, cefepime, amikacin, meropenem, teicoplanin and vancomycin. 

In order to clarify, the following alteration was made in the manuscript: “Of the multidrug-resistant bacteria isolated, one (7.1%) was vancomycin-resistant Enterococcus faecalis isolated from a culture of a skin secretion sample and 13 (92.9%) were carbapenem-resistant and aminoglycoside-susceptible K. pneumoniae isolated from cultures of urine samples. Most of these latter patients (69,2%) received microbiologically appropriate treatment with either amikacin or gentamicin and 77,8% had an outcome of cure or clinical improvement.”

3) Mortality was quite low. Was it associated with the infectious disease? The same description should be given regarding cause of hospital admission, if possible.

Tables 3 and 4 show the association of admission and death to the following infectious diagnosis: pneumonia, urinary infection and skin/soft tissue infection, none of which showed statistical significance in the bivariate analysys. 

4) “The authors found that infection with such strains did not represent a risk factor for mortality or hospital admission” This statement should be better discussed. I suggest to describe more deeply the type of infection caused by MDR bacteria treated with outpatient treatment.

The following information was added to the manuscript: “The authors found that infection with such strains did not represent a risk factor for mortality or hospital admission, differing from what has been reported in other studies that evaluated those outcomes in patients with multidrug-resistant bacterial infection. 19,20 This difference might be explained by the fact that in this study, most patients with multidrug-resistant isolates had urinary tract infections caused by K. pneumoniae strains that were susceptible to aminoglycosides, which were available for targeted short-term treatment in OPAT.”

---

## [Decision Letter · Decision Letter 1]

19 Oct 2020

Factors associated with successful completion of outpatient parenteral antibiotic therapy in an area with a high prevalence of multidrug-resistant bacteria: 30-day hospital admission and mortality rates

PONE-D-20-18208R1

Dear Dr. Thais Cristina Garbelini Salles

We’re pleased to inform you that your manuscript has been judged scientifically suitable for publication and will be formally accepted for publication once it meets all outstanding technical requirements.

Kind regards,

Francesco Di Gennaro

Academic Editor

PLOS ONE

Additional Editor Comments (optional):

Dear Authors congratulations

Reviewers' comments:

Reviewer's Responses to Questions

**Comments to the Author**

1. If the authors have adequately addressed your comments raised in a previous round of review and you feel that this manuscript is now acceptable for publication, you may indicate that here to bypass the “Comments to the Author” section, enter your conflict of interest statement in the “Confidential to Editor” section, and submit your "Accept" recommendation.

Reviewer #1: All comments have been addressed

Reviewer #2: All comments have been addressed

2. Is the manuscript technically sound, and do the data support the conclusions?

Reviewer #1: Yes

Reviewer #2: Yes

3. Has the statistical analysis been performed appropriately and rigorously? 

Reviewer #1: Yes

Reviewer #2: Yes

4. Have the authors made all data underlying the findings in their manuscript fully available?

Reviewer #1: Yes

Reviewer #2: Yes

5. Is the manuscript presented in an intelligible fashion and written in standard English?

Reviewer #1: Yes

Reviewer #2: Yes

6. Review Comments to the Author

Reviewer #1: (No Response)

Reviewer #2: (No Response)

7. PLOS authors have the option to publish the peer review history of their article (what does this mean?). If published, this will include your full peer review and any attached files.

Reviewer #1: No

Reviewer #2: No

---

## [Editor Report · Acceptance letter]

9 Nov 2020

PONE-D-20-18208R1 

Factors associated with successful completion of outpatient parenteral antibiotic therapy in an area with a high prevalence of multidrug-resistant bacteria: 30-day hospital admission and mortality rates 

Dear Dr. Salles:

I'm pleased to inform you that your manuscript has been deemed suitable for publication in PLOS ONE. Congratulations! Your manuscript is now with our production department. 

Kind regards, 

on behalf of

Dr. Francesco Di Gennaro 

Academic Editor

PLOS ONE